# Novel 2-Hydroxypropyltrimethyl Ammonium Chitosan Derivatives: Synthesis, Characterization, Moisture Absorption and Retention Properties

**DOI:** 10.3390/molecules26144238

**Published:** 2021-07-12

**Authors:** Yingqi Mi, Qin Miao, Jingmin Cui, Wenqiang Tan, Zhanyong Guo

**Affiliations:** 1Key Laboratory of Coastal Biology and Bioresource Utilization, Yantai Institute of Coastal Zone Research, Chinese Academy of Sciences, Yantai 264003, China; yqmi@yic.ac.cn (Y.M.); qmiao@yic.ac.cn (Q.M.); jmcui@yic.ac.cn (J.C.); wqtan@yic.ac.cn (W.T.); 2Center for Ocean Mega-Science, Chinese Academy of Sciences, 7 Nanhai Road, Qingdao 266071, China; 3University of Chinese Academy of Sciences, Beijing 100049, China

**Keywords:** chitosan, moisture absorption, moisture retention

## Abstract

Recent years have seen a steady increase in interest and demand for the use of humectants based on biodegradable natural polymers in many fields. The aim of this paper is to investigate the moisture absorption and retention properties of 2-hydroxypropyltrimethyl ammonium chitosan derivatives which were modified by anionic compounds via ion exchange. FTIR, ^1^H NMR, and ^13^C NMR spectroscopy were used to demonstrate the specific structures of chitosan derivatives. The degrees of substitution for objective products were calculated by the integral ratio of hydrogen atoms according to ^1^H NMR spectroscopy. Meanwhile, moisture absorption of specimens was assayed in a desiccator at different relative humidity (RH: 43% and 81%), and all target products exhibited enhanced moisture absorption. Furthermore, moisture retention measurement at different relative humidity (RH: 43%, 81%, and drier silica gel) was estimated, and all target products possessed obviously improved moisture retention property. Specifically, after 48 h later, the moisture retention property of HACBA at 81% RH was 372.34%, which was much higher than HA (180.04%). The present study provided a novel method to synthesize chitosan derivatives with significantly improved moisture absorption and retention properties that would serve as potential humectants in biomedical, food, medicine, and cosmetics fields.

## 1. Introduction

Humectants are compounds which can obtain and retain moisture from the humid air to achieve moisturizing and moistening effects [1,2,3]. Humectants have been widely used in food, cosmetics, and other fields due to their water stabilization capacity [4]. Specifically, on the one hand, it is crucial to reduce the relative humidity level and keep moisture stability in order to maintain the sensory attributes and extend shelf life in foods [4,5]. On the other hand, for human skin, it is of vital importance to increase hydration with the aid of humectants [6,7]. Moreover, humectants can serve as carriers to deliver vitamins, antioxidants, and anti-inflammatory drugs to skin [6,7]. It is reported that the humectants industry is growing rapidly, and the market will reach a value of $26.27 billion by 2022 [8]. As common humectants, propylene glycol and glycerin possess strong moisture absorption property but relative weak moisture retention property [1,4]. As for hyaluronic acid (HA), it is an important functional ingredient in the cosmetics field due to its strong moisture-retention ability [9,10]. However, the extensive use of HA is limited by its high price. In recent years, with obvious improvements in consumer consciousness in cosmetics and food safety, seeking competitive natural polysaccharides and their derivatives with moisture absorption and retention properties to serve as humectants in the cosmetics and food industry has drawn more and more attention.

Chitosan, β-(1–4)-2-amino-2-deoxy-d-glucan, as a derivative synthesized by the chemical deacetylation of natural polysaccharide chitin, is the second most abundant natural polysaccharide [11,12]. In recent years, there has been a sustained growth in public interest and demand for chitosan due to its renewable and non-toxic properties, especially in the biomaterials, cosmetics, food, and other fields, reflected by numerous articles [13,14]. Particularly, many outstanding properties such as good moisture absorption and retention properties, biocompatibility, and biodegradability, allow its wide use as potential humectant in the food and cosmetics fields [10,15]. However, the large-scale industrial applications of pristine chitosan are incomparable to that of synthetic biopolymers because of relatively weak bioactivity and poor water solubility in neutral and alkaline pH. Fortunately, the hydroxyl (OH) and amino (NH_2_) groups in chitosan chains provide effective reactive sites in order to introduce functional groups and develop novel chitosan derivatives through chemical modification, with a dramatic increase in functional properties.

2-hydroxypropyltrimethyl ammonium chloride chitosan (HACC) is a kind of polycationic compound with good water solubility, moisture absorption, and biocompatibility [16,17,18]. Although the moisture absorption and retention properties of HACC are better than those of chitosan, it is still not enough to be widely used as a humectant. Meanwhile, it is reported that carboxymethyl chitosan (CMCS) is not only soluble in water, but possesses unique properties, including low toxicity, biocompatibility, high-activity, and good moisture absorption and retention properties, which make it an attractive option, widely used in the food and cosmetics fields [19,20]. It is worth noting that CMCS has a similar skeletal structure to HA, but the moisture absorption and retention properties of CMCS have received more attention. Based on the above, we hypothesize that the extension space of the molecule will enlarge due to the repulsion of the negative charge of COOH when COOH is introduced to chitosan molecular chains [19,21]. Combined with the hydrophilicity of COOH, the ability of the derivatives to bind water will be significantly higher than that of chitosan, thus enhancing the moisture absorption and retention properties.

The aim of this study is to prepare chitosan derivatives with high moisture absorption and retention properties, which can be used as competitive humectants in the food and cosmetics fields. Therefore, five kinds of HACC derivative bearing compounds containing carboxylic anions were synthesized via ion exchange: carboxymethyl chitosan anion-HACC conjugates (HACCMC), carboxymethyl inulin anion-HACC conjugates (HACCMI), pyrrolidone carboxylate anion-HACC conjugates (HACPC), lactate anion-HACC conjugates (HACLC), and betaine anion-HACC conjugates (HACBA), respectively. FTIR, ^1^H NMR and ^13^C NMR spectroscopy were used to demonstrate the specific structures of chitosan derivatives. The moisture absorption and retention properties of HACC derivatives were estimated at different relative humidity conditions.

## 2. Results and Discussion

### 2.1. Chemical Synthesis and Characterization

The 2-hydroxypropyltrimethyl ammonium chitosan derivatives were synthesized by multistep reactions (Scheme 1). FTIR, ^1^H NMR, ^13^C NMR spectroscopy were used to demonstrate the specific structures of chitosan derivatives.

#### 2.1.1. Fourier Transform Infrared (FTIR) Spectra

Figure 1 shows the FTIR spectra of chitosan, inulin, and their derivatives, respectively. For chitosan, the asymmetric broad band observed at 3500–3300 cm^−^^1^ is due to OH vibrations, the characteristic absorbance band observed at 2925 cm^−^^1^ is attributed to the CH stretching vibration of methylene, the absorption band appearing at 1624 cm^−^^1^ belongs to vibration modes of amide I, and the absorption bands ranging from 1250 to 850 cm^−1^ belong to the glycosidic ring [22]. In particular, the characteristic absorbance band appeared at 1155 cm^−^^1^, consistent with the glycosidic linkage. Compared with chitosan, after carboxymethylation, the FTIR spectra of CMCS appears with a new characteristic peak at 1732 cm^−1^, which can be assigned to the COOH stretching vibration. A similar pattern is found in the spectra of CMIL [23]. For CMIL, in addition to the characteristic absorbance bands appearing at 3410 cm^−^^1^ (OH stretching vibration), 2935 cm^−^^1^ (CH stretching vibration), and 1034 cm^−1^ (C-O stretching vibration), the new peak at 1745 cm^−1^ shows the stretching vibration of COOH [24]. As for HACC, after quaternization, the new characteristic peak at 1489 cm^−^^1^ illustrates the successful introduction of trimethylammonium group. As for the five kinds of 2-hydroxypropyltrimethyl ammonium chitosan derivatives prepared by ion exchange, the same characteristic peak appears at about 1478 cm^−1^, which can be assigned to the stretching vibration of N^+^(CH_3_)_3_ [3]. In particular, for HACCMC and HACCMI, the new absorption peaks appear at 1413 and 1415 cm^−1^, which belong to the absorption vibration of negative carboxylic acid ions. In the spectra of HACBA, after the introduction of betaine anion, new peaks appear at 1149 cm^−1^ and 1031 cm^−1^, which are assigned to the characteristic peaks of C-N stretching vibration [18]. Therefore, the specific structures of chitosan, inulin, and their derivatives were preliminarily concluded by FTIR spectra.

#### 2.1.2. Nuclear Magnetic Resonance (NMR) Spectra

The ^1^H NMR spectra of chitosan, inulin, and their derivatives are shown in Figure 2. The chemical shifts of unmodified chitosan are shown as follows: peaks at 4.51 ppm, 3.54–4.05 ppm, and 3.10 ppm are attributed to the hydrogen protons of [H1], [H3]–[H6], and [H2], respectively [25]. Compared with chitosan, after carboxymethylation, the ^1^H NMR spectra of CMCS appears as a new chemical shift at 4.37 ppm, which can be assigned to the methylene protons of CH_2_COOH [23]. For CMIL, compared to inulin, in addition to the chemical shifts appearing at 3.24–5.45 ppm ([H1]–[H6]), and 5.36 ppm (an anomeric proton in H1-Glc), the new peak at 4.53 ppm shows the introduction of the methylene protons of CH_2_COOH, proving the successful synthesis of carboxymethyl inulin [24]. In ^1^H NMR spectrum of HACC, the signal of trimethyl ammonium protons of N^+^(CH_3_)_3_ group appears at 3.14 ppm, which further proves the successful introduction of the trimethyl ammonium group. Besides, the new chemical shifts are easy to observe for HACC in the spectra 4.21 ppm (b), 2.45 ppm (c), and 2.84 ppm, attributed to -CH_2_ (a). As for the five kinds of 2-hydroxypropyltrimethyl ammonium chitosan derivatives prepared by ion exchange, the same characteristic shift appears at about 3.14 ppm, which can be assigned to the hydrogen protons of N^+^(CH_3_)_3_ [3]. This means that the methylene ammonium group still exists after the reaction. For HACCMC, compared to HACC, the new chemical shift is located at 3.94 ppm, which is assigned to protons of methylene. The same chemical shift appears in the spectrum of HACCMI. The new chemical shifts of HACPC appearing at 4.07 ppm (e) and 2.28 ppm (f, g) are attributed to the resonance of pyrrolidone carboxylate. For HACBA, the signal of the N^+^(CH_3_)_3_ group of betaine appears at 3.16 ppm in the ^1^H NMR spectra, which further demonstrates the successful introduction of betaine anion [3,26]. Besides, the structures of chitosan, inulin, and their derivatives are also confirmed by ^13^C NMR spectra. In Figure 3, the specific peaks of carbons are marked clearly. It is obvious that the chemical shifts of chitosan are mainly distributed in the range of 50–100 ppm. After carboxymethylation, a new peak at 174 ppm is clearly observed in the spectra of CMCS which can be assigned to the resonance of carbons of COOH. The same signal is observed in the spectrum of CMIL. For HACC, the signal of carbons of N^+^(CH_3_)_3_ group appears at 54.26 ppm, which further proves the successful introduction of trimethyl ammonium group. As for conjugates of anionic compounds and HACC, compared to HACC, new chemical shifts appearing at δ 177, δ 160, δ 181, δ 182, and δ 169 ppm are attributed to carbons of COO^−^ groups in HACCMC, HACCMI, HACPC, HACLC, and HACBA [27]. So, the specific structures of chitosan, inulin, and their derivatives were further demonstrated by the ^1^H NMR and ^13^C NMR spectra.

### 2.2. Degrees of Substitution (DS)

The DS of chitosan and inulin derivatives were calculated in accordance with the ^1^H NMR spectra. For chitosan derivatives, the DS were measured by using the integration of [H1] as an integral standard peak [24]. As for inulin derivatives, the integration of [H4] was used as an integral standard peak. For CMCS, the specific calculation method is listed in the materials and methods section, and the integral values of hydrogen protons of ^1^H NMR spectra are shown in Figure 4. The DS of CMCS, CMIL, HACC, HACCMC, HACCMI, HACPC, HACLC, and HACBA was determined as 72.00%, 46.00%, 81.00%, 51.00%, 52.00%, 68.00%, 58.00%, and 53.00%, respectively (Table 1). In addition, the material yield of each step is the quality yield. In particular, yield = (actual quality/theoretical quality) × 100%.

### 2.3. Moisture Absorption Assay

Chitosan, inulin, and their derivatives were compared with the control (Sodium Hyaluronate, HA) for water absorption. Figure 5 shows the moisture absorption of all tested samples at relative humidity of 81% and 43%, respectively. From Figure 5a it is easy to observe that the moisture absorption exhibits a time-dependent rise in all tested samples and achieves a maximum value at 48 h. Chitosan and inulin possess low moisture absorption compared to their derivatives and HA at 48 h. To be specific, firstly, compared to pure chitosan and inulin, CMCS and CMIL possess much more moisture absorption of 49.14%, and 43.56%, respectively. Secondly, for all samples, with the passage of time, moisture absorption increases gradually. For instance, the moisture absorption of HACCMI is 11.66%, 56.14%, 68.06%, 82.86, and 95.27%, when the test time is 4, 12, 32, 40 and 48 h. Thirdly, the order of moisture absorption for all tested samples is as follows: HACCMI > HACBA > HACCMC > HACLC > CMCS > CMIL > HACPC > HA > inulin > HACC > chitosan. Obviously, for HACC, after the introduction of anionic compounds, HACC derivatives have a more significant moisture absorption property. Among all the derivatives, HACCMI has the strongest moisture absorption property, with a hygroscopicity of up to 95.27% when the test time is 48 h. The moisture absorption of all tested samples at a relative humidity of 43% is shown in Figure 5b. From the figure, similar conclusions are summarized as follows: firstly, it is found that the moisture absorption has a positive correlation with tested time. Secondly, all HACC derivatives have good moisture absorption. Compared to HA with an index of 32.36% at 48 h, the moisture absorption indices of HACCMC, HACCMI, HACPC, HACLC, and HACBA are 74.16%, 94.20%, 35.56%, 50.66%, and 79.83%, respectively.

The results in Figure 5 show that HACC derivatives had much stronger moisture absorption at relative humidity 81% and 43%. In brief, the moisture absorption of all tested samples was ranked in this order: HACCMI > HACBA > HACCMC > HACLC > CMCS > CMIL > HACPC > HA > inulin > HACC > chitosan. Specifically, some rules can be drawn from the figure: firstly, after carboxymethylation, both CMCS and CMIL had significantly improved moisture absorption. This was attributed to the introduction of carboxyl groups (COOH). The polar groups (COOH) on the surface of molecules were easy to form hydrogen bonds with water [19,20]. Secondly, the moisture absorption presented a time-dependent rise and achieves a maximum value at 48 h. Itis worth noting that the moisture absorption index increased rapidly during the first few hours. This was because hydrogen bonds between water and COOH heightened. At high relative humidity conditions (RH 81% and 43%), water is easy to access the surface of the chitosan residues and there are easy hydrogen-bond interactions with the chitosan molecular chains [19]. As time goes on, hydrogen bond formation becomes saturated, so the rate of increase in the moisture absorption index slows down. Finally, all HACC derivatives possessed enhanced moisture absorption, and the moisture absorption capacity of some derivatives such as HACCMC, HACCMI, and HACBA is even more than that of HA. The intermolecular hydrogen bonds increased with the augmentation of polar groups, which helps the improvement in moisture absorption [20].

### 2.4. Moisture Retention Assay

Chitosan, inulin, and their derivatives were compared with the control (Sodium Hyaluronate, HA) for moisture retention. Figure 6 shows the moisture retention of all tested samples at relative humidity of 81%, 43%, and drier silica gel, respectively. From Figure 6a, it is observed that all tested samples possessed moisture retention in a concentration dependent manner. A gradual increase in the moisture retention property of derivatives is found as the time increases from 2 to 48 h. Compared with the weak moisture retention of 156.78% of HACC, the introduction of anionic compounds improves moisture retention, and the values of moisture retention of anionic compounds modified HACC (HACCMC, HACCMI, HACLC, HACPC, HACBA) are 264.81%, 290.89%, 272.89%, 259.40%, and 372.34% at maximum test time of 48 h, respectively. A similar pattern is found in Figure 6b. The results revealed that the moisture retention at relative humidity 43% is significantly enhanced with the passage of time. At the same test time, the order of moisture retention capacity is ranked as HACBA  >  HACCMI > HACLC  >  HACPC > HACCMC > HA > HACC. Obviously, after the introduction of anionic compounds like betaine, pyrrolidone carboxylate, CMIL, and so on, the derivatives possess significantly enhanced moisture retention. Among these, betaine modified HACC (HACBA) had the highest moisture retention capacity (372.34%), much higher than that of HA (169.08%) at 48 h. The moisture retention of the tested sample in the drier silica gel is shown in Figure 6c. In this assay, HA was used as the positive control. Similarly, CS and HACC possess weak moisture retention, and when test time is 48 h their moisture retention are only 25.49% and 27.14%, respectively. After ion exchange, compared to CS and HACC, five kinds of HACC derivatives show relatively strong moisture retention. In addition, the order of moisture retention is ranked as follows: HACBA > HACCMI > HACLC > HACPC > HACCMC > HA > HACC.

The results from Figure 6 show that HACC derivatives had much stronger moisture absorption at relative humidity of 81% and 43%. In short, taking DS into account, the moisture retention of chitosan derivatives ranked in the order: HACBA > HACCMI > HACLC > HACPC > HACCMC > CMIL > IL > CMCS > HA > HACC > CS. To be specific, some conclusions can be drawn from Figure 6: firstly, the chitosan derivatives possessed dramatic moisture retention at every test time, and the conclusion substantiated that the introduction of anionic compounds including carboxymethyl polysaccharide anion or other anionic compounds greatly enhanced the moisture retention of chitosan derivatives [20,21]. Secondly, HACBA had the best moisture retention property at all relative humidity conditions, much stronger than that of HA. HACBA, with good moisture retention, was attributed to the active groups on the surface of molecules including COOH, OH, and N^+^(CH_3_)_3_ [19,28]. These active groups could form hydrogen bonds with water. Moreover, with the augmentation of intermolecular hydrogen bonds, the molecular chains could bond large amounts of water and retain them in a spacious network. To sum up, the HACC derivatives including HACCMI, HACBA, HACCMC, HACLC, and HACPC represented better moisture absorption and retention than that of HA, and has potential use as a moisture retention ingredient in food, medicine, and cosmetics fields.

## 3. Materials and Methods

### 3.1. Materials

The pristine chitosan was purchased from Qingdao Baicheng Biochemical Corp. (Qingdao, China). Its average molecular weight (MW) and degree of deacetylation are 200 kDa and 85%, respectively. Inulin which was extracted from chicory with average DP of 40 was supplied by Bai Chuan Biotechnology Co., Ltd. (Leqing, China). Sodium hydroxide, ammonium sulfate, sodium carbonate, isopropyl alcohol, chloroacetic acid, acetone, ethanol, and hydrochloric acid were provided by Sinopharm Chemical Reagent Co., Ltd. (Shanghai, China). The materials including sodium hyaluronate, betaine, sodium pyrrolidone-carboxylate, sodium lactate, and 3-chloro-2-hydroxypropyltrimethyl ammonium chloride were purchased from Sigma-Aldrich Chemical Corp. (Shanghai, China). All the chemical reagents and solvents were of analytical grade and used as received.

### 3.2. Chemical Synthesis

#### 3.2.1. Preparation of Carboxymethyl Chitosan (CMCS) and Carboxymethyl Inulin (CMIL)

CMCS was prepared as in the previous method [23]. Firstly, chitosan (1.61 g) was dispersed in 15 mL of isopropanol at 25 °C. After dispersing evenly, 4.03 mL of sodium hydroxide solution (40%, *w*/*v*) was added dropwise. Then the reaction solution was stirred in a metal bath at 50 °C for alkalization. After 1 h, 37.03 mL of chloroacetic acid solution (10%, *w*/*v*) was dropped in and the mixture stirred for 4 h. Finally, CMCS was obtained by precipitation, filtration, washing, and drying to constant weight.

CMIL was prepared as in the previous method [24]. In brief, inulin (1.62 g) was dispersed in 15 mL of isopropanol at 25 °C. After dispersing equably, 4.03 mL of sodium hydroxide solution (40%, *w*/*v*) was added drop by drop. After 1 h stirring in a metal bath at 50 °C, chloroacetic acid solution (20 mL, 10%) was dropped. Four hours later, CMIL was obtained by precipitation with excessive acetone, filtration, washing with acetone, and drying to constant weight.

#### 3.2.2. Preparation of 2-Hydroxypropyl Trimethyl Ammonium Chloride Chitosan (HACC)

In a representative procedure for the preparation of HACC [3], chitosan solution was obtained by dissolving chitosan (1.61 g, 10 mmol) in 80 mL of isopropanol with vigorous stirring. The solution was reacted at 60 °C ahead of adding 4 mL of sodium hydroxide solution (40%, *w*/*v*). Four hours later, 3-chloro-2-hydroxypropyl trimethyl ammonium chloride solution (12.5 mL, 60%) was added into the front solution with vigorous stirring for 10 h at 80 °C. The right amount of hydrochloric acid was added in order to adjust the pH of the reaction system to 7. HACC was synthesized by precipitation, rinsing, and drying in vacuo.

#### 3.2.3. Preparation of 2-Hydroxypropyl Trimethyl Ammonium Chitosan Derivatives

To obtain 2-hydroxypropyl trimethyl ammonium chitosan derivatives, taking HACCMC as an example, 1.25 g (5 mmol) CMIL was accurately weighed and dissolved in 20 mL of deionized water. After stirring at room temperature for 2 h, 10 mL of HACC solution (10%, *w*/*v*) was added into the front reaction mixture drop by drop. After the reaction at room temperature for 12 h, the solution was dialyzed (molecular weight cutoff: 500, MD77, Lombard, IL, USA) against deionized water for 48 h. HACCMC was obtained by lyophilizing in a vacuum freeze dryer.

### 3.3. Analytical Methods

#### 3.3.1. Fourier Transform Infrared (FTIR) Spectroscopy

The FTIR spectra were performed on a Jasco-4100 FTIR spectrometer (Tokyo, Japan, provided by JASCO China Co. Ltd., Shanghai, China). The measurement was taken in the 4000–400 cm^−1^ region at a resolution of 4.0 cm^−1^. The tested samples were scanned 16 times using the KBr disks at 25 °C.

#### 3.3.2. Nuclear Magnetic Resonance (NMR) Spectroscopy

The ^1^H NMR spectra and ^13^C NMR spectra were carried out on a Bruker Avance III 500 NMR Spectrometer (500 MHz, Fällanden, Switzerland, provided by Bruker Tech. and Serv. Co., Ltd., Beijing, China) at 25 °C. All the prepared samples were dissolved in D_2_O.

The degrees of substitution (DS) of target products were measured according to ^1^H NMR spectra [24]. For example, the DS of CMCS were carried out by the integral ratio of the hydrogen atom bonded to Carbon 1 of the chitosan and hydrogen atoms of carboxymethyl group. The formula is shown as follows:(1)$$\mathrm{DS}\;(\%)=\frac{\frac{I_{H_{7},\;CMCS}}{2}}{I_{H_{1,\;CS}}}$$
where $I_{H_{7},\;CMCS}$ is the integral values of the H_7_ of CMCS (4.5–4.6 ppm); 2 is the number of protons in H_7_ of CMCS; $I_{H_{1,\;CS}}$ is the integral value of the hydrogen atom bonded to Carbon 1 of chitosan backbone (3.9–4.1 ppm).

### 3.4. Moisture Absorption Measurement

Moisture absorption measurement was performed following the previous reported method with slight modifications [29]. The tested samples were ground and dried to constant weight. Initially, 0.5 g test specimen (*W*_0_) was accurately weighed. Then, all the tested specimens were placed in a hermetically sealed container at different relative humidity (RH = 43%, 81%) for 48 h. Finally, according to predetermined intervals, tested samples were taken out of the container and dried by wiping with a dustless cloth in order to obtain the final accurate quality (*W*_1_). The moisture absorption was measured according to the formula:(2)$$\mathrm{Moisture}\mathrm{absorption}\;(\%)=\frac{W_{1}-W_{0}}{W_{0}}\times 100$$

### 3.5. Moisture Retention Measurement

Moisture retention measurement was performed in accordance with the previous reported method [4]. The tested samples were ground and dried to constant weight. Initially, 0.5 g test specimen with water content of 10% was placed in in a hermetically sealed dryer at different relative humidity (43%, 81%, and drier silica gel). Then, according to predetermined intervals, tested samples were taken out of the container and dried by wiping dustless cloth in order to obtain the final accurate quality of residual water. The moisture retention was measured by the following formula:(3)$$\mathrm{Moisture}\mathrm{retention}\;(\%)=\frac{H_{1}}{H_{0}}\times 100$$
where *H*_0_ represents the weight of water in the initial samples, and *H*_1_ is the weight of water in sample after putting in the silica gel.

### 3.6. Statistical Analysis

Every experiment was repeated three times. The experimental results were reported as mean ± standard deviation and analyzed with one-way analysis of variance (ANOVA). Significant difference analysis was determined in accordance with Scheffe’s multiple range test. The level of *p* < 0.05 was determined statistically significant.

## 4. Conclusions

In the present study, anionic compounds including CMCS, CMIL, betaine, sodium pyrrolidone carboxylate, and sodium lactate were successfully introduced to cationic compound (HACC) through ion exchange in water. The FTIR, ^1^H NMR and ^13^C NMR spectroscopy analysis verified the specific structures of chitosan derivatives. The DS of chitosan and inulin derivatives were calculated in accordance with the ^1^H NMR spectra. Meanwhile, moisture absorption of all samples was assayed at different relative humidity (RH: 43% and 81%). Five kinds of HACC derivatives possessed dramatic moisture absorption at all times, and the conclusion illustrated that the introduction of anionic compounds including carboxymethyl polysaccharide anion or other anionic compounds greatly enhanced the moisture absorption of chitosan derivatives. It is worth noting that HACCMI had the best moisture absorption property at all relative humidity conditions, much stronger than that of HA. Moreover, moisture retention tests at different relative humidity (RH: 43%, 81%, and drier silica gel) were estimated, and all target products possessed obviously improved moisture retention property. Therefore, the present work can provide an attractive method to synthesize chitosan derivatives with obviously improved moisture absorption and retention properties that can serve as potential humectants in the biomedical, food, medicine, and cosmetics fields.

## Data Availability

All data contained in the manuscript are available from the authors.

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
