# Peer review of "Novel 2-Hydroxypropyltrimethyl Ammonium Chitosan Derivatives: Synthesis, Characterization, Moisture Absorption and Retention Properties"

_molecules, 2021, doi:10.3390/molecules26144238_

Round 1

Reviewer 1 Report

This MS reports about the investigaton of moisture absorption and retention properties of 2-hydroxypropyltrimethyl ammonium chitosan derivativatives that can be used for as competitive humectants for food and cosmetics. Compounds were characterized via FTIR and H1 NMR before estimation of adsorption via relative humidity.

The paper is well written and the subject is clearly presented. I therefore recommend publication of this paper in the present form.

Author Response

This MS reports about the investigaton of moisture absorption and retention properties of 2-hydroxypropyltrimethyl ammonium chitosan derivativatives that can be used for as competitive humectants for food and cosmetics. Compounds were characterized via FTIR and H1 NMR before estimation of adsorption via relative humidity.

The paper is well written and the subject is clearly presented. I therefore recommend publication of this paper in the present form

Answer: Thank you for your approval.

Reviewer 2 Report

The authors achieved a series of chemical modification of chitosan in order to study their moisture absorption and retention properties with application if the food and cosmetic fields.

The originality of the paper relies mainly on the application. The authors compared their products with hyaluronic acids and obtained improved properties, which deserves to be published.

Minor comments:

  • Line 123: in the sentence « hydrogen protons of CH2COOH », I suppose that the authors mean the methylene protons and not the acidic protons. It is not written.
  • How did the authors attribute the signals a and c of the product HAAC? Is the attribution confirmed by 2D NMR and any other data?
  • Line 188: “It's worth noting that the moisture absorption index increased rapidly during the first few hours. This was because hydrogen bonds between water and COOH weakened over time [19]. The explanation is not clear, and I don’t follow the reason of a weakening of the H bonds with time. Besides, an entropic penalty, coming from water absorption and interaction through H-bonds, should also play in important role, which is not mentioned in the discussion. Finally, I was wondering if degradation of modified chitosan can take place during the absorption experience, which could have an impact of the absorption profile.

Reviewer 3 Report

The manuscript of the article entitled "Novel 2-Hydroxypropyltrimethyl Ammonium Chitosan Derivatives: Synthesis, Characterization, Moisture Absorption and Retention Properties" by authors Yingqi Mi, Qin Miao, Jingmin Cui Wenqiang Tan and Zhanyong Guo is mostly well presented and should be interesting for readers of Molecules. I have following remarks: The alkylation of chitosan using choroacetate in the presence sodium hydroxide should take place mainly on the NH2 groups, which should be included to discussion and also in the Scheme 1. (See the case of alkylation ethanolamine by choroacetate described e.g. in Bioorg. Med. Chem. Lett. 12, 3085, 2002). The relevant discussions and conclusions cannot be given on the base of IR spectra which have too low resolution. Please add full IR spectra with better resolution. For better characterization of chitosan derivatives should be convenient to use 13C NMR spectroscopy. The present manuscript should be acceptable after major revision for publishing in the Molecules.

Reviewer 4 Report

The paper is concerning the preparation of new chitosan derivatives as water absorbers. It is extremely interesting for the intriguing perspectives of application in different fields. Thus, the paper could be accepted but, in my opinion, with revisions.
1) the syntheses of all the HA and chitosan derivatives studied in the paper would be added, together with more detailed experimental data, such as the yields.
2) also the characterization would be implemented, in particular, with some data concerning the molecular weight, achieved for instance through MALDI analyses, to see if the treatments with NaOH and/or HCl, cited in the text, can break the polymeric chain, thus strongly changing the structure of the derivatives in comparison with the starting polymers.
3) a study carried out with TD NMR methods to observe the water behaviour in these materials could offer interesting information concerning the chemical-physical properties of the materials.
4) a study of the protonic NMR spectra after/during water absorption coud give further informations, in particular explain why the order of moisture absorption is not the same of that of moisture retention. conformational changes occur in the polymers? 
5) a TGA/DSC study could give useful data, too.
In conclusion, in my opinion the water behaviour in these materials would be more deeply explored.

Round 2

Reviewer 3 Report

The revised manuscript is acceptable in present form.